# A Mysterious Asian Firefly Genus, *Oculogryphus* Jeng, Engel & Yang (Coleoptera, Lampyridae): The First Complete Mitochondrial Genome and Its Phylogenetic Implications

**DOI:** 10.3390/insects15070464

**Published:** 2024-06-21

**Authors:** Yu-Xia Yang, Ya Kang, Xue-Ying Ge, Shuai-Long Yuan, Xue-Yan Li, Hao-Yu Liu

**Affiliations:** 1The Key Laboratory of Zoological Systematics and Application, School of Life Science, Institute of Life Science and Green Development, Hebei University, Baoding 071002, China; yxyang@hbu.edu.cn (Y.-X.Y.); 20208017081@stumail.hbu.edu.cn (Y.K.); gexueying@stumail.hbu.edu.cn (X.-Y.G.); 20238017078@stumail.hbu.edu.cn (S.-L.Y.); 2Hebei Basic Science Center for Biotic Interaction, Hebei University, Baoding 071002, China; 3State Key Laboratory of Genetic Resources and Evolution, Kunming Institute of Zoology, Chinese Academy of Sciences, Kunming 650223, China

**Keywords:** firefly, mitochondrial gene rearrangement, intergenic spacer, phylogeny

## Abstract

**Simple Summary:**

The family Lampyridae, commonly known as fireflies, is a cosmopolitan group comprising approximately 100 genera and 2200 species. *Oculogryphus* Jeng, Engel & Yang, 2007 is a small firefly genus endemic to Asia with only four known species and the phylogenetic position of it remains unresolved. It has been considered enigmatic because of its characteristic morphology intermingling different subfamilies, while its systematic status has never been rigorously tested due to a lack of molecular data. In this study, we obtained a large series of *O*. *chenghoiyanae* Yiu & Jeng, 2018 from southeastern China. We successfully sequenced and annotated its complete mitochondrial genome and clarified the systematic status of *Oculogryphus*.

**Abstract:**

The firefly genus *Oculogryphus* Jeng, Engel & Yang, 2007 is a rare-species group endemic to Asia. Since its establishment, its position has been controversial but never rigorously tested. To address this perplexing issue, we are the first to present the complete mitochondrial sequence of *Oculogryphus*, using the material of *O. chenghoiyanae* Yiu & Jeng, 2018 determined through a comprehensive morphological identification. Our analyses demonstrate that its mitogenome exhibits similar characteristics to that of *Stenocladius*, including a rearranged gene order between *trnC* and *trnW*, and a long intergenic spacer (702 bp) between the two rearranged genes, within which six remnants (29 bp) of *trnW* were identified. Further, we incorporated this sequence into phylogenetic analyses of Lampyridae based on different molecular markers and datasets using ML and BI analyses. The results consistently place *Oculogryphus* within the same clade as *Stenocladius* in all topologies, and the gene rearrangement is a synapomorphy for this clade. It suggests that *Oculogryphus* should be classified together with *Stenocladius* in the subfamily Ototretinae at the moment. This study provides molecular evidence confirming the close relationship between *Oculogryphus* and *Stenocladius* and discovers a new phylogenetic marker helpful in clarifying the monophyly of Ototretinae, which also sheds a new light on firefly evolution.

## 1. Introduction

The family Lampyridae, also known as fireflies, is a cosmopolitan group within the superfamily Elateroidea [1,2,3,4]. There are approximately 100 genera and 2200 species [5], and many more are yet to be discovered. The higher-level classification of fireflies has changed significantly over time [6,7,8,9,10], and recent phylogenetic efforts have provided deeper insight into their classification by expanding morphological datasets [11,12,13], molecular datasets [14,15,16], or both [17]. However, some genera within Lampyridae remain unsolved in their phylogenetic positions [16].

One of the *incertae sedis* groups of Lampyridae is *Oculogryphus* Jeng, Engel & Yang, 2007 [18]. The genus was erected based on *O. fulvus* Jeng, 2007, located in Vietnam, by monotypic and original designation. Subsequently, three more species were added from Vietnam and China [19,20,21]. This genus is morphologically distinctive with males having large compound eyes that are significantly emarginate posteriorly and approximate ventrally [18,20], while females are neotenic and completely larviform [21]. Originally, *Oculogryphus* was thought to be a mysterious taxon with features intermingling those of Rhagophthalmidae, Luciolinae, Lampyrinae, or the Ototretine–Ototretadriline complex [18]. It was later revealed to be closely related to the ototretine genus *Stenocladius* Fairmaire in Deyrolle and Fairmaire, 1878 based on a morphology-based phylogenetic study of Lampyridae [13,19]. However, Janisova and Bocakova [22] did not consider it as a member of Ototretinae when they synonymized the subfamily Ototretadrilinae Crowson, 1972 with the latter and redefined the limits of the group based on adult morphology. After that, with further discoveries of additional *Oculogryphus* species, some additional evidence was found to support its close relationship to *Stenocladius* in the morphological characters of males and neotenic females, as well as bioluminescence behavior for attracting mates [20,21]. Despite these great efforts, the systematic status of *Oculogryphus* remains controversial at the moment and is placed as Lampyridae *incertae sedis* in the latest classification [16]. When there remains controversy over the taxonomic status of some morphospecies, it becomes necessary to use molecular data to test phylogenetic relationships [23,24,25]. However, no molecular data has been available for *Oculogryphus* until now, thereby preventing us from rigorously testing its phylogenetic position.

During our recent study, we conducted extensive insect collections in southeastern China. Fortunately, we were able to obtain a large series of *Oculogryphus* materials among these samples. The availability of these valuable materials provided us with the opportunity to investigate its phylogenetic position based on the molecular data of Lampyridae. Mitochondrial genomes (mitogenomes) and nuclear ribosomal DNA (rDNA) repeat units are widely used in insect phylogenies [26,27]. Furthermore, alongside raw sequence data for phylogenetic comparison [28], the sequences and structural characteristics of mitogenomes, including the secondary structure of RNA genes, base content, and gene arrangement, can reflect differences in the functions and evolutionary patterns of diverse taxa [29,30]. As the number of mitogenome reference sequences has increased, comparative feature analysis among and within certain groups has become more informative [30]. Gene rearrangement is one of the most frequently investigated characteristics of metazoan mitogenomes [30,31,32,33,34,35], and it has been an extremely useful phylogenetic marker [31,36,37]. The gene rearrangement between *trnC* and *trnW* was recently detected in *Stenocladius* [38], prompting us to thoroughly explore whether similar characteristics exist in *Oculogryphus* or not. Moreover, three regions of the nuclear rDNA repeat unit (e.g., 18S, 5.8S, and 28S) are classically conserved molecular markers with very low rates of genetic saturation (defined as multiple substitutions at the same site) [39], which are commonly used to construct phylogenies in insects [15,40,41,42].

In the present study, we will analyze the characteristics of the mitochondrial genome of *Oculogryphus* for the first time and include it in the reconstruction of Lampyridae’s molecular phylogeny. Our aims are to (1) make a comparison between *Oculogryphus* and *Stenocladius* in the mitochondrial genes arrangement, (2) preliminarily investigate the phylogenetic position of *Oculogryphus* using a comprehensive molecular dataset following Chen et al. [15], and (3) further investigate the phylogenetic position of *Oculogryphus* with all representative taxa with complete mitogenomes. Through these results, we will test the hypothesis of whether *Oculogryphus* is a sister to *Stenocladius* or not.

## 2. Materials and Methods

### 2.1. Materials

We conducted extensive insect collections in southeastern China from 2020 to 2021 and obtained a large series of *Oculogryphus* materials among these samples. Some *Oculogryphus* materials were located from Nanling National Natural Reserve, Guangdong Province, China, 24.911547–24.937461° N, 113.011181–113.045847° E, 826–1278 m, 26.V–15.VI.2020–2021, H.D. Yang & J.B. Tong leg. The remaining were from Jiulianshan, Jiangxi Province, China, 24.537250–24.587017° N, 114.452717–114.466114° E, 416–644 m, 27.V–17.VI.2020, H.D. Yang leg. The materials were preserved in 100% ethanol at −20 °C for long-term storage prior to DNA extraction. All of them were deposited at the Museum of Hebei University, Baoding, China (MHBU). The material for the molecular study was attached with a voucher number (MHBU, 3CA0034).

### 2.2. Morphological Technique

The specimens were initially softened in water, followed by dissection of the genitalia and hind wings. Subsequently, the male genitalia were cleared in a 10% NaOH solution, examined and photographed in glycerol, and finally affixed onto a paper card for permanent preservation. The left hind wing of each specimen was removed from the body and mounted in neutral balsam between a microscope slide and a cover slip. At least one specimen was dissected for each locality, with additional specimens treated if any damage occurred during dissection. Images of the genitalia and hind wings were captured using a Leica M205A stereo microscope. Multiple layers were stacked using Helicon Focus 7. Post-processing of images was conducted using Adobe Photoshop 2020.

The distribution information was gathered from the original publications [21] and the newly collected material of the present study.

### 2.3. DNA Extraction, Mitochondrial Genome Sequencing, and Assembly

Total genomic DNA was extracted from chest muscle using a DNeasy Blood and Tissue kit (QIAGEN, Beijing, China), following the manufacturer’s protocol, and the extracted DNA was stored at −20 °C for further molecular studies. Libraries were constructed and sequenced using the Nova Seq 6000 platform (Illumina, Alameda, CA, USA) at Berry Genomics (Beijing, China). Paired-end reads with 150 bp length and an insert size of 350 bp were sequenced. Approximately 6G of data were produced for the sample to ensure the minimum sequencing coverage of 10×. The mitogenome was assembled using NOVOPlasty version 3.8.3 [43], with variable K-mer sizes (21, 24, 27, 30, 33, 36), until a consistent result was achieved.

### 2.4. Genome Annotation and Sequence Analyses

Gene annotation was performed using Geneious 2019.2 software [44] with invertebrate mt code and then checked by manual proofreading according to its related species [45], with *Stenocladius bicoloripes* serving as the reference sequence [38]. The secondary structures of twenty-two tRNAs were predicted using the MITOS Web Server (http://mitos.bioinf.uni-leipzig.de/index.py, assessed on 20 January 2024) [46] and subsequently verified by the tRNA Scan-SE server (http://lowelab.ucsc.edu/tRNAscan-SE/, assessed on 20 January 2024) [47,48]. The skewnesses were determined based on the base composition of nucleotide sequences using the formula: AT skew = [A − T]/[A + T], and GC skew = [G − C]/[G + C] [49]. The non-synonymous substitutions (Ka) were calculated in DnaSP [50] based on the 13 PCGs, with those of *Rhagophthalmus ohbai* as a reference [51]. The mitogenome map was produced using the OrganellarGenomeDRAW Server visualization tool (http://ogdraw.mpimp-golm.mpg.de/index.shtml, assessed on 20 January 2024) [52]. Additionally, the Tandem Repeat Finder program (https://tandem.bu.edu/trf/trf.html, assessed on 20 January 2024) [53] was utilized to predict tandem repeats within non-coding regions. The sequence generated in this study has been deposited in GenBank with accession number (ON985401).

### 2.5. Dataset Assembly and Phylogenetic Analyses

In our study, we analyzed three different datasets including *Oculogryphus*. First, we investigated the position of this genus within Lampyridae using the most comprehensive DNA datasets (dataset-1: both nuclear and mitochondrial genes, including 10 mitochondrial genes: COX1-2, ATP6&8, ND1, ND4-5, CYTB, 12S, 16S, and three rDNA regions: 18S, 5.8S, and 28S by Chen et al. [15]. In this analysis, a total of 53 Lampyridae species served as ingroups, and eight species of Elateroidea were used as outgroups due to the inconsistency of the sister group [16] (Appendix A). Second, because not all molecular markers of the representative species are available for us in this analysis, which may influence the estimation of the phylogenetic relationships, we created another two datasets that included only complete mitogenomes (dataset-2: PCGRNA matrix, including 13 PCGs and two rRNAs; and dataset-3: PCG12RNA matrix, including the 1st and 2nd codon positions of the 13 PCGs and two rRNAs). In this second analysis, a total of 43 species of Lampyridae were chosen as the ingroups, and eight species of Elateroidea were also used as outgroups (Appendix A). Except for the newly sequenced mitogenome and three rDNA regions (18S, 5.8S, and 28S) of *Oculogryphus*, all other sequences were downloaded from GenBank [54].

The individual genes of PCGs and nuclear sequences were aligned using the MAFFT algorithm [55] implemented in PhyloSuite [56] with the L-INS-I strategy. The alignments were trimmed using G-blocks [57] and then concatenated into the aforementioned different datasets. The pre-defined partitions for different datasets follow a consistent pattern, with protein-coding genes divided according to codon positions, while other genes are each placed into a separate partition. PartitionFinder [58] and ModelFinder [59] were utilized to search for the optimal partitioning scheme and models for each dataset. Maximum likelihood (ML) trees were constructed using IQ-Tree version 2.0.7 [60] with 1000 SH-aLRT replicates. Bayesian inference (BI) analyses were conducted in MrBayes v3.2.6 [61] until the average standard deviation of splitting frequency was less than 0.01. Additionally, the second datasets were analyzed under the default CAT+GTR model using PhyloBayes-MPI version 1.8 [62]. Two independent MCMC chains were run until they satisfactorily converged (maxdiff < 0.3). Interactive Tree of Life (iTOL, http://itol.embl.de, assessed on 25 January 2024) [63] was used to visualize the phylogenetic tree.

## 3. Results

### 3.1. Taxonomy

Class Insecta Linnaeus, 1758Order Coleoptera Linnaeus, 1758Family Lampyridae Rafinesque, 1815Genus *Oculogryphus* Jeng, Engel & Yang, 2007*Oculogryphus chenghoiyanae* Yiu & Jeng, 2018*Oculogryphus chenghoiyanae* Yiu & Jeng, 2018 [21]: 67.Figure 1 and Figure 2

Examined material. CHINA, Guangdong, Nanling National Natural Reserve: 1♂, 1023 m, 113.025572° N, 24.910075° E, 15.VI.2020, J.B. Tong leg. (Figure 2H); 1♂, 1048 m, 113.025733° N, 24.909214° E, 27.V.2020, J.B. Tong leg.; 2♂, 1056 m, 113.025628° N, 24.908536° E, 13.VI.2021, H.D. Yang leg. (Figure 1B and Figure 2F,G); 1♂, 1278 m, 113.011181° N, 24.937461° E, 26.–28.V.2021, H.D. Yang leg.; 1♂, 1100 m, 113.024792° N, 24.931478° E, 30.V–1.VI.2020, J.B. Tong leg.; 1♂, 846 m, 113.040283° N, 24.914000° E, 26–28.V.2021, H.D. Yang leg.; 1♂, 844 m, 113.042678° N, 24.912725° E, 30.V–1.VI.2021, H.D. Yang leg. (Figure 2A); 1♂, 826 m, 113.045847° N, 24.911547° E, 29–30.V.2021, H.D. Yang leg. Jiangxi, Jiulianshan: 1♂, 620 m, 114.464547° N, 24.538331° E, 27.V.2020, H.D. Yang leg. (Figure 1C and Figure 2B); 1♂, 644 m, 114.463817° N, 24.538936° E, 9.VI.2020, H.D. Yang leg.; 1♂, 615 m, 114.466114° N, 24.537464° E, 27.V.2020, H.D. Yang leg. (Figure 1A and Figure 2C); 1♂, 633 m, 114.4654442° N, 24.537250° E, 27.V.2020, H.D. Yang leg.; 2 ♂, 643 m, 114.464667° N, 24.537661° E, 17.VI.2020, H.D. Yang leg. (Figure 1D and Figure 2D,E); 1♂, 416 m, 114.447947° N, 24.584364° E, 27.V.2020, H.D. Yang leg.; 1♂, 406 m, 114.452717° N, 24.587017° E, 1.VI.2020, H.D. Yang leg.

Distribution. China (Jiangxi, Guangdong, Hong Kong).

Descriptive notes. In the hind wing of the male (Figure 1), the MP_3+4_ is vestigial, with its trace being more or less clear along the whole length (e.g., Figure 1A,B). Sometimes it is branched terminally into MP_3_ and MP_4_, respectively (e.g., Figure 1C,D).

In the aedeagus (Figure 2), the median lobe is thinned apically in the dorsal view (Figure 2A–H), and strongly curved dorsally in the lateral view, which looks nearly even in width along the whole length and feebly expanded at the apex (Figure 2a–h). The parameres are elongate and narrowly rounded at apices, but they can never reach the apex of the median lobe (Figure 2A–H). They are usually diverging from each other apically, rarely nearly parallel (Figure 2F). The basal piece is roughly a U-shaped band (Figure 2A–H), sometimes presenting with a more or less distinct median notch at the caudal margin (e.g., Figure 2B,E–G). It is shorter than (e.g., Figure 2D,F–H) or approximately as long as (e.g., Figure 2A–C,E) the median lobe.

Remarks. Considering the rarity, or little current understanding of this taxon, we conducted a thorough and comprehensive morphological observation to ensure accurate specific identification.

The external morphology of this species was thoroughly described in the original manuscript [21]. Since there is little variation within these materials, it is unnecessary to reiterate the descriptions of the same characters herein.

### 3.2. General Features of the Mitochondrial Genome

The complete mitochondrial genome of *O. chenghoiyanae* is 16,546 bp in size, with a nucleotide composition biased towards A and T (47. 1% for A, 33.4% for T, 12.1% for C, and 7.4% for G). It has a positive AT-skew (0.17) and a negative GC-skew (−0.24) (Appendix A). In addition, the value of non-synonymous substitutions is 0.20957 (Appendix A). The mitogenome is a typical double-strand circular molecule containing 13 protein-coding genes (PCGs), 22 transfer RNA genes (tRNAs), 2 ribosomal RNA genes (rRNAs), and an A+T-rich region, of which 14 genes (8 tRNAs, 4 PCGs, and 2 rRNAs) were transcribed from the minority strand (N-strand) while the others (14 tRNAs and 9 PCGs) were encoded on the majority strand (J-strand) (Figure 3a, Appendix A). Gene arrangement differs from most fireflies, e.g., as in [64,65,66] (Figure 3b, as *ND2*-*trnW*-*trnC*-*trnY*), with a rearrangement between *trnC* and *trnW* genes observed in *O. chenghoiyanae* (as *ND2*-*trnC*-*trnW*-*trnY*). In addition, all tRNA genes fold into the standard clover-leaf structure except *trnS1*, which is missing the dihydrouridine (DHU) arm (Appendix A). Moreover, there are 12 gene overlaps ranging from 1 to 8 bp in length, as well as five intergenic spacer regions between genes, with the longest between *trnC* and *trnW* (702 bp) (Appendix A). Notably, the intergenic spacer between *trnC* and *trnW* genes contains six tandemly repeated units (TRU, 111 bp) along with a partial 44 bp repeat unit. Each TRU is composed of two parts (Figure 3b). One part corresponds to a portion of the nucleotide sequence of *trnW* (29 bp), which is thought to be a remnant of *trnW* and relatively conserved except for only one base mutation from A to G in the fifth repeat unit. The other part of the TRU is 82 bp, which is identical to one another.

### 3.3. Phylogenetic Analyses

Using the ML and BI analyses, almost congruent topologies were produced for the mitochondrial and nuclear genes (Figure 4a–d, Appendix A). The monophyly of Ototretinae was not recovered. With the exception of *Brachylampis blaisdelli* (located in North America), all other representative taxa (from the Oriental region) were split into two different clades. One clade was composed of *Drilaster* Kiesenwetter, 1879, *Lamellipalpus* Maulik, 1921, *Flabellototreta* Pic, 1911, *Mimophaeopterus* Pic, 1930, and *Ceylonidrilus* Pic, 1929 (BS = 100, PP = 0.999), which was the sister to the remaining Lampyridae species (BS = 72, PP = 0.969). The other clade consisted of *Oculogryphus*, *Stenocladius*, *Ototretadrilus* Pic, 1921, and an unidentified species of Ototretinae (BS = 100, PP = 1), which was placed in a distant position from the proceeding clade. Within the latter clade, *Oculogryphus* was recovered as a sister to *Stenocladius*, but albeit with low supporting values (BS = 52, PP = 0.58).

Like the above, the phylogenetic analyses of the complete mitogenomes (concatenated into two different datasets) showed that Ototretinae was a paraphyly, consisting of two branches (Figure 4e and Appendix A). One branch included only *Drilaster* (BS = 100, PP = 1), and the other grouped with *Oculogryphus* and *Stenocladius* (BS = 100, PP = 1). *Drilaster* was always in the basal clade sister to the rest of the Lampyridae (BS = 98–100, PP = 1). And *Oculogryphus* and *Stenocladius* were further sisters to the others (PP = 1) using the BI analysis, while they were grouped with either Lampyrinae or Luciolinae under the ML analysis albeit with lower supporting values (BS = 38–59).

## 4. Discussion

### 4.1. Species Identification

We identified all materials as *O. chenghoiyanae* following the key by Yiu & Jeng [21]. A comprehensive study of these available materials indicates some variations in their aedeagi and hind wings, which are important for identifying the *Oculogryphus* species [18,19,20,21].

In the original description of *O. chenghoiyanae* ([21]: Figure 2), the hind wing venation is similar to some of our material (Figure 1A), in which the MP_3+4_ is vestigial, and its vein trace is disrupted. But in some other cases, the MP_3+4_ vein trace is complete (Figure 1B), or may also be bifurcate (Figure 1C,D). Despite this variability, the vestigial nature of MP_3+4_ is consistent across specimens of *O. chenghoiyanae*, unlike all others of *Oculogryphus* having well-developed ones ([18]: Figure 8; [19]: Figure 3; [20]: Figure 3).

Also, the aedeagus of *O. chenghoiyanae* is variable in the shapes of its component structures. As originally described ([21]: Figure 4A), the basal piece is usually a U-shaped band in the dorsal view (e.g., Figure 2A,C,D), sometimes with a median notch at the caudal margin (e.g., Figure 2B,E,H). It is always bent dorsally, and abruptly (e.g., Figure 2a,c,f,h; [21]: Figure 4B) or progressively (e.g., Figure 2b,d,e) narrowed basally in the lateral view. Parameres are generally elongate and tapered apically, but their length varies. Some are short, less than half the length of the median lobe (e.g., Figure 2B,G), while others are relatively long, approximately 2/3 the length of the median lobe (e.g., Figure 2F,H), and the rest fall between these lengths. The median lobe varies in stoutness in the dorsal view (e.g., Figure 2A,D; [21]: Figure 4A), with some being slender (e.g., Figure 2B,E). However, its shape is quite consistent in the lateral view (Figure 2a–h). While the male genitalia are generally considered to be the most stable and reliable structure in insect taxonomy e.g., [67], their shape may vary more or less within the species [68]. This is not an individual case of fireflies, such as *Microphotus* LeConte, 1866 [69], but it typically exhibits only quantitative variation. Therefore, it is essential to thoroughly examine a large number of individuals when describing a new species, in order to comprehensively and accurately recognize its defining characteristics.

Although there are some variations within *O. chenghoiyanae*, its males could be well differentiated from all other *Oculogryphus* species by a combination of the following characters: a pronotum and elytra of highly contrasting colors, with a reddish-brown pronotum, and black elytra except for brown at the humeri; the elytra are widest near the middle part, with a narrow epipleura; a hind wing with vestigial MP_3+4_; an aedeagus with parameres less than 2/3 the length of the median lobe, which is uniform in width along the whole length in the lateral view. The consistency of these characters, coupled with their close geographical distribution (or the sympatric range), leads us to conclude that all specimens should be classified as *O. chenghoiyanae*.

### 4.2. Comparisons of the Mitogenomes between Oculogryphus and Stenocladius

With *O. chenghoiyanae* as the studied material, this study presented the first complete mitogenome record for *Oculogryphus*. Similar to the mitogenomes of *Stenocladius* [38], a rearrangement between the *trnW* and *trnC* was detected in *Oculogryphus*, with a long intergenic spacer existing between the two rearranged genes. Within this long intergenic spacer, some remnant genes of *trnW* were also found (Figure 3a). However, there are some detailed differences between *Oculogryphus* and *Stenocladius* in the intergenic spacer between the *trnW* and *trnC* genes (Appendix A). This non-coding region of *Oculogryphus* (702 bp) is much longer than that of *Stenocladius* (241 to 376 bp) [38]. Additionally, the remnant gene of *trnW* in *Oculogryphus* is longer (29 bp) and repeated more times (six) than that of *Stenocladius* (23 bp, one or two repeats) [38]. Furthermore, the sequences between the remnant genes are identical to one another (82 bp) in *Oculogryphus*, while they are different in *Stenocladius* [38]. From another perspective, the intergenic spacer of *Oculogryphus* is composed of six TRUs, while only repeat sequence units (RSUs) or remnant genes exist in *Stenocladius*.

The presence of the intergenic spacers located in positions involved in rearrangements, combined with the presence of gene remnants that changed positions within intergenic spacers, provides evidence for identifying plausible rearrangement pathways [70,71]. Similar to what occurred in all *Stenocladius* mitogenomes [38], the rearrangement of *trnW* and *trnC* in *Oculogryphus* also occurred through transposition, with the genes moving to different places on the same strand [71]. Transposition can be explained by a tandem duplication and random loss model (TDRL) [72,73]. TDRL involves a tandem duplication of a continuous segment of genes such that the original segment and its copy are placed consecutively, followed by the loss of one copy of each redundant gene [74]. Further assumptions regarding rearrangement processes were made based on the principle of parsimony for *Oculogryphus* (Figure 3b). First, gene duplication of the gene cluster *trnW*-*trnC* with intergenic spacer in ancestral order resulted in a tandemly repeated structure with repeated duplication: (*trnW*-*trnC*)-(*trnW*-*trnC*)_6_-(*trnW*-*trnC*). Subsequently, a random loss of genes occurred, including the upstream *trnW* gene, middle gene cluster (*trnW*-*trnC*)_6_, and downstream *trnC*. Finally, a new gene order was produced: *trnC*-*trnW* with an intergenic spacer consisting of six tandemly repeated units.

What is particularly noteworthy is that numerous studies have demonstrated a correlation between high rates of mitochondrial gene rearrangement and elevated nucleotide replacement rates in animal lineages [75]. The non-synonymous substitutions (Ka) serve as a valuable parameter for assessing the evolutionary rate [76]. Indeed, the Ka values for *Oculogryphus* and *Stenocladius* exceed 0.2, significantly surpassing those of *Drilaster* within Ototretinae (Appendix A), suggesting that a high rate of nucleotide substitution may be one contributing factor to gene rearrangement. However, it should be noted that higher Ka values (>0.2) do not necessarily result in gene rearrangement, as evidenced by Lampyrinae and Photurinae (Appendix A).

### 4.3. Phylogenetic Position of Oculogryphus

In the present study, we reconstructed the phylogeny of Lampyridae using molecular data, including *Oculogryphus* in the analyses for the first time. Our results indicate that the monophyly of Ototretinae has never been recovered, consistent with the recent molecular phylogenetic analyses [15,38,65]. The clade consisting of *Oculogryphus* and *Stenocladius* always appeared separate from *Drilaster* and/or some related genera. Or this clade was grouped together with *Ototretadrilus* when the latter was included in the comprehensive molecular phylogeny (Figure 4a–d). While this result aligns with most molecular phylogenetic analyses [15], it contradicts morphological classification such as Janisova and Bocakova’s work [22], in which *Oculogryphus* is not considered a member of Ototretinae, and *Stenocladius* along with *Brachypterodrilus* Pic, 1918, *Baolacus* Pic, 1915, and *Falsophaeopterus* Pic, 1911 belongs to an ototretine subgroup characterized by less prominent lateroposterior angles of the pronotum. In contrast, *Drilaster* and *Ototretadrilus* are included in the other subgroup exhibiting contrasting characteristics.

A single rearrangement occurred in the stem lineage leading to all *Stenocladius* mitogenomes and *Oculogryphus* (i.e., not a parallel acquisition). Specifically, at least a rearrangement between the *trnW* and *trnC* genes is a synapomorphy supporting the sister relationship of *Oculogryphus* and *Stenocladius*, but this was not found in the mitogenome of *Drilaster* [38]. This finding is in line with Jeng’s (2008) phylogenetic analysis based on morphological data, as well as the opinions held by Jeng and his coauthors [19,20,21]. In light of this discovery, *Oculogryphus* should theoretically be classified in the same category as *Stenocladius* within the subfamily Ototretinae [16] at the moment.

Unfortunately, not all genera sampling or all molecular markers of Ototretinae are available in the phylogenetic analyses. It is particularly regrettable that the complete mitochondrial genome of *Ototretadrilus* has been unavailable until now. Therefore, the monophyly of Ototretinae cannot be clarified herein. Additionally, the systematic status of Ototretadrilinae (probably at least included *Ototretadrilus*, *Stenocladius,* and *Oculogryphus*) remains unclear. The determination of whether *Oculogryphus* is a member of true Ototretinae or Ototretadrilinae can only be made once the monophyly of the newly redefined Ototretinae [22] has been clarified. This requires more samplers representing all related ototretine genera (a total of 21 genera hitherto known) [21] to be included in future phylogenetic analysis.

No matter how, the gene rearrangement between *trnW* and *trnC* may serve as a valuable phylogenetic marker for testing the monophyly of Ototretinae, which is currently unsatisfactory in its definition. Thus, it is strongly encouraged that more mitogenomes are accumulated for Ototretinae and that extensive comparative analysis of their gene order is conducted, in order to improve classification and gain deeper insight into firefly evolution, which may be driven by the gene rearrangement.

## 5. Conclusions

In the present study, we have successfully sequenced and annotated the complete mitochondrial genome for the rare-species firefly genus *Oculogryphus* for the first time. The studied material of *O. chenghoiyanae* was determined through a comprehensive morphological identification. Our analysis showed that the characteristics of this sequence are similar to those of *Stenocladius*, including a rearranged gene order between *trnW* and *trnC*, as well as a large intergenic spacer (702 bp) between the two rearranged genes, within which six remnants of *trnW* (29 bp) exist. Furthermore, our phylogenetic reconstruction of Lampyridae incorporating this sequence in the cladistics analyses indicated that *Oculogryphus* is consistently grouped with *Stenocladius* and that gene rearrangement is a synapomorphy for this monophyletic clade. Based on these results, we suggest *Oculogryphus* be placed in Ototretinae at the moment. Nevertheless, further sequencing and analysis of additional genera within Ototretinae are needed to verify these obtained results and clarify the monophyly of the subfamily. This will also help us better understand the evolution of fireflies in general.

## Figures and Tables

**Figure 1 insects-15-00464-f001:**
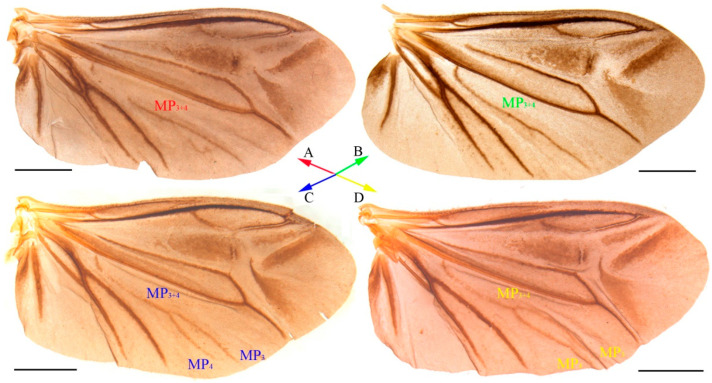
Hind wings of *Oculogryphus chenghoiyanae* Yiu & Jeng, 2018, unfolded views. Scale bars: 1.0 mm. Different colors denote different material from different localities or on collecting dates to show variations within species: (**A**,**C**,**D**): specimens from Jiulianshan, Jiangxi, China; (**B**): specimens from Nanling National Natural Reserve, Guangdong, China (detailed information denoted in Examined Material part).

**Figure 2 insects-15-00464-f002:**
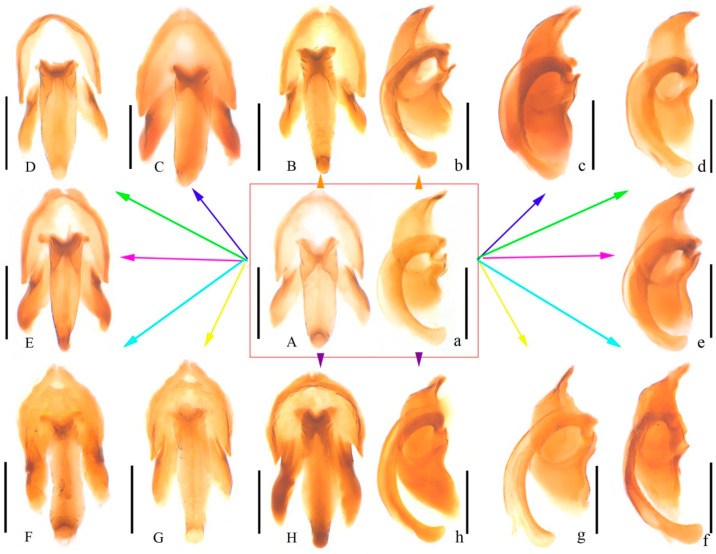
Aedeagi of *Oculogryphus chenghoiyanae* Yiu & Jeng, 2018: (**A**–**H**) dorsal views; (**a**–**h**) lateral views. Scale bars: 0.5 mm. Different colors denote different material from different localities or on collecting dates to show variations within species: (**A**,**F**–**H**) (**a**,**f**–**h**): specimens from Jiulianshan, Jiangxi, China; (**B**–**E**,**b**–**e**): specimens from Nanling National Natural Reserve, Guangdong, China (detailed information denoted in Examined Material part).

**Figure 3 insects-15-00464-f003:**
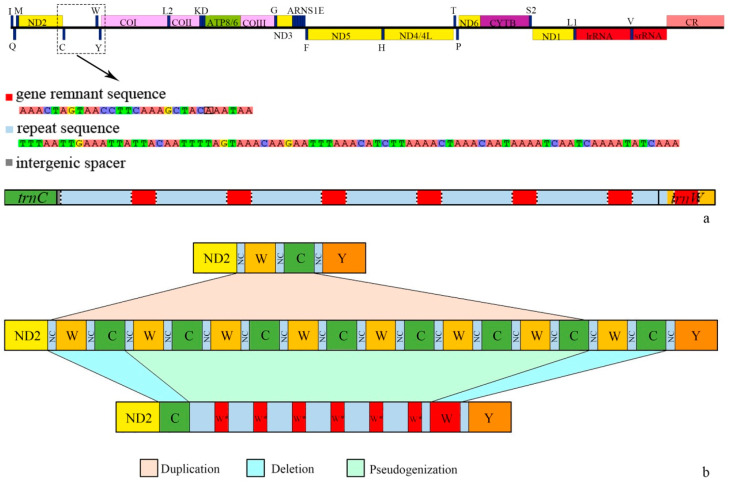
(**a**) A linear map of the complete mitogenome of *Oculogryphus chenghoiyanae* and the intergenic spacer between *trnW* and *trnC*, showing the number, position, and nucleotide sequence of the gene remnant. (**b**) The putative mechanism of gene rearrangement. The size of the genes is not proportional. W* indicates a 29 bp *trnW* residue; W, C and Y are *trnW*, *trnC* and *trnY*, respectively; ND2 is the protein-coning gene *ND2*; NC means intergenic spacer.

**Figure 4 insects-15-00464-f004:**
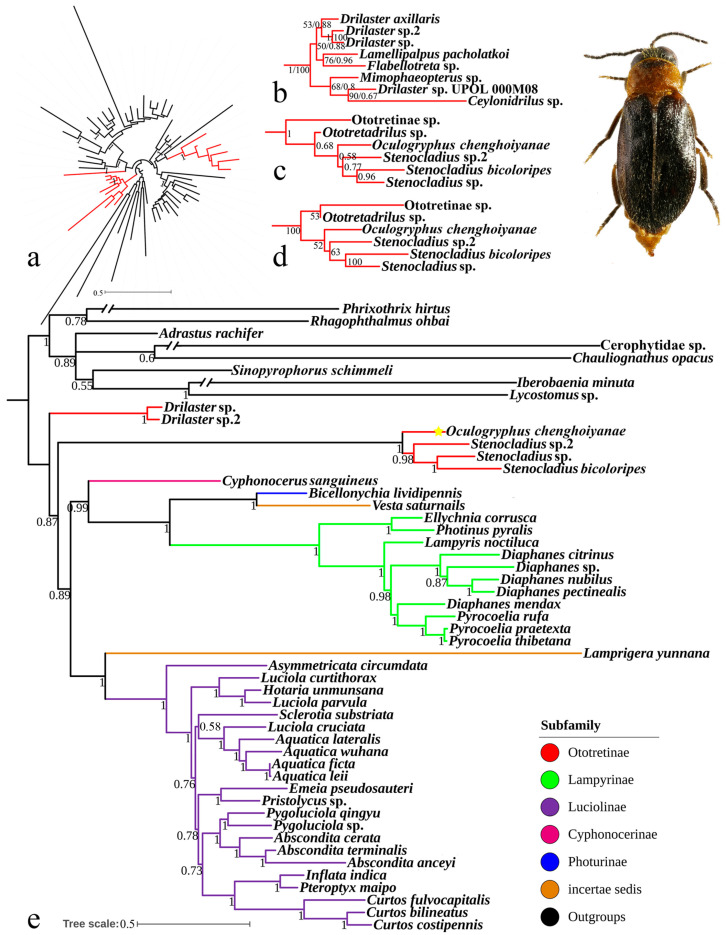
(**a**) Bayesian inference phylogenetic tree of Lampyridae inferred based on ten mitogenome and three nuclear markers. Ototretinae is highlighted in red. (**b**–**d**) Taxon names and statistical support for branches of a; only Ototrtinae clade shown (except *Brachylampis blaisdelli*), see full trees in Appendix A. (**e**) Bayesian inference phylogenetic tree of Lampyridae based on PCGRNA with heterogeneous models CAT+GTR in PhyloBayes. The yellow star denotes the phylogenetic position of *Oculogryphus chenghoiyanae*.

## Data Availability

The newly sequenced mitochondrial genome in this study has been uploaded to GenBank (ON985401).

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
