# Peer review of "A Mysterious Asian Firefly Genus, Oculogryphus Jeng, Engel & Yang (Coleoptera, Lampyridae): The First Complete Mitochondrial Genome and Its Phylogenetic Implications"

_insects, 2024, doi:10.3390/insects15070464_

Round 1
Reviewer 1 Report
Comments and Suggestions for Authors
The manuscript insects-3053212, submitted to insects MDPI, entitled “A mysterious Asian firefly genus, Oculogryphus Jeng, Engel & Yang (Coleoptera, Lampyridae): First complete mitochondrial genome and its phylogenetic implications” carried interesting results and well written. The research of the manuscript is based on both techniques (morphological and molecular) to explore the species taxonomy. Some suggested changes are mentioned in comments portion to revise the draft for improvement.
Morphological technique: reference is missing for the technique, please add suitable reference for the methodologies
Some abbreviations are missing the details, please add detail of abbreviations where first used or add abbreviations details in separate section for better understanding to the readers. Please recheck throughout the manuscript
However, it is worth noting that the monophyly of Ototretinae has never been recovered, consistent with the recent molecular phylogenetic analyses [15, 38, 65]. The clade Oculogryphus + Stenocladius always appeared separate from Drilaster and or some related genera, but together grouped with Ototretadrilus when included in the comprehensive molecular phylogeny (Figure 5a–d).” The sentence is too lengthy please rephrase the sentence
Gaps of the study are not clearly highlighted however it will be interesting if the gaps of the study will be incorporated more clearly and future research to overcome these
Conclusion: The conclusion portion may be more precise in writing.
References: Please write the references in the text and the reference portion according to the author's instructions and Journal style/formatting.
Please double-check for typos and inconsistencies in Journal style/formatting/author instructions as among others, double spaces, spellings of the words, English vocabulary, missing italics, scientific names, missing information

Minor corrections especially in rephrasing of sentences
Author Response
Please see the response in the attachment, thanks.

Reviewer 2 Report
Comments and Suggestions for Authors
Authors sequenced the complete mt genome of one rare firefly, reconstructed its phylogenetic position and discussed its placement by comparing gene rearrangement of mt genomes. The overall paper is good and reasonable quality. I have two major concerns:
1. What is the pre-partitioned scheme? Are the datasets been partitioned by gene, codon position, or both combined? It should be clearly stated in the methods section.
2. This study tries to resolve a deep relationship within the family that the third codon position of mt genes is often saturated and only provides phylogenetic noise. Have you tested the impact of the third codon position?
Not very relevant to this study, but since you have found stable variation from different populations, have you sequenced their coi gene and seen if there are large genetic gaps between different populations? By saying this, I want to know the potential of cryptic species.
Comments on the Quality of English LanguageThe English is clear but some parts are not concise enough.
For example, I suggest to modify your simple summary as:
Simple Summary: The family Lampyridae, commonly known as fireflies, is a cosmopolitan group comprising approximately 100 genera and 2200 species. Oculogryphus Jeng, Engel & Yang, 2007 is a small firefly genus endemic to Asia with only four known species and the phylogenetic position of it remains unresolved. It has been considered enigmatic because of its characteristic morphology intermingling different subfamilies, while its systematic status has never been rigorously tested due to a lack of molecular data. In this study, we obtained a large series of O. chenghoiyanae Yiu & Jeng, 2018 from southeastern China. We sequenced and annotated its complete mitochondrial genome and clarified the systematic status of Oculogryphus.
Author Response

(The authors gave the same response as above.)

Reviewer 3 Report
Comments and Suggestions for Authors
Dr. Yang and Collaborators here present the complete mitochondrial genome of O. chenghoiyanae, a member of genus Oculogtyphus that is currently classified as incertae sedis, and they conduct a phylogenetic analysis to place this latter in the context of Lampyridae.
The study is conducted using methods that are standard for the field and is in line – in general terms - with other studies published in this same journal. I do not see any major flaw, and therefore I am suggesting ‘minor revisions’. I am nevertheless requesting some extra clarifications on the methods and to revise some passage of the discussion section for clarity, as well as to realign results and discussion in some parts.
Line 32: O. c. is defined as ‘representative’. Does it simply mean that the authors, in the current study, use this species to study the phylogenetic placement of the genus, or there is some reason why this species is somehow special, and ‘representative’ within the genus?
Line 42: ‘a new .. marker .. to clarify the monophyly of Ototretinae’. Considering that Ototretinae are not recovered as monophyletic, this statement is somehow misleading.
Line 76, 105 (and elsewhere): the sampling strategy is not clear. Did the Authors sample O. chenghoiyanae on purpose? Or did they collect insects (or lampyrids) at large, realized that there was O. chenghoiyanae among their samples, and then decided to go for mitogenome sequencing based on opportunity?
Paragraph 2.3: please provide more details on the sequencing (library type, mean and stability of coverage on the reconstructed sequence).
Paragraph 2.3: the assembly is described as stemming from a simple NOVOPLASTY analysis. Based on experience it is rarely this simple, especially given the presence of a long repeat region that can complicate the assembly when short reads are used. Did all NP runs (different k) produce circularized assemblies? Were they all identical in sequence? If not, how was one assembly preferred over others? The term ‘until the best results were achieved’ is unsatisfactory in this respect.
Line 143: Ka/Ks calculations require an outgroup or an alignment to be calculated, while at this point of the presentation only one sequence is available/has been introduced. Please rephrase.
Line 149: the sequence of the mitochondrial genome is not currently visible in NCBI. To my knowledge, releasing the sequence before publication is not a requirement of the journal, and therefore I am not complaining. Nevertheless I wish to point out that a manuscript whose main result is the sequence of a complete mitochondrial genome is expected to be accompanied by a quality NCBI genome record. Please confirm, in the response letter, that the mitogenome has been correctly accessioned, include all annotation and it is has not been flagged as ‘unverified’ during the NCBI revision stage.
Paragraph 2.5: this is slightly confusing to read, as the three datasets are introduced in the first subparagraph and then described in the second subparagraph. Please reword for clarity.
Paragraph 2.5: furthermore, considering that dataset 1 includes nuclear ribosomal genes as well, it is not clear how this information was acquired for the species under study. Were ribosomal sequences assembled from the raw data and included in the analysis? If YES, please describe the assembly/extraction procedure. If NOT, then I do not understand why dataset 1 exists at all, as it would be a subset of datasets 2/3. In this case, unless there is a significant reason to include dataset 1, this could be deleted altogether. In any case please report if datasets are complete (i.e. all gene sequences are present for the focal species as well as for all other species).
Line 169: if I read correctly, partition finder was used to identify the best fitting model for the dataset, bu this was imposed as a single partition. Noteworthy, PartitionFinder was designed to identify the best partitioning scheme AND the best model for each partition, not just the latter. Considering the centrality of the phylogenetic reconstruction for the manuscript, and considering that the phylogenetic trees presented are not unproblematic, I suggest a proper partitioning strategy is conducted.
Line 185: reference is made to Fig.1-11. Of which paper? If it is not the current one, which is the rationale? Please clarify.
Paragraphs 3.1 and 4.1: this layout is typical of morphology/taxonomy manuscripts that describe new species, while it is not common of molecular phylogeny papers. More typical of a molecular phylogenetic paper would be ‘samples of O. chenghoiyanae were collected in … and were identified according to …’. This, obviously, assumes that species identification is not critical and not questionable.
Considering the rarity, or little current understanding, of this taxon, and considering that a careful morphological observations were indeed conducted, I am not suggesting that this part is deleted. Nevertheless it should find its proper integration in the manuscript: a) physical position in the text; b) formatting and subparagraph structure in line with the rest, c) role in the overall argumentation. More specifically on this latter, I think the role of this part is to convince the reader that the sequence is really from O. chenghoiyanae, and not from a similar species. This part may be reworded with this in mind.
Fig. 1: In the absence of (minimal) geographic information, the image is difficult to read (I assume very few people can guess from the coordinates). The three disconnected points provide a very limited account of the species distribution, to the point that the figure could be safely removed. Can the Authors refer to the literature to identify/hypothesize the boundaries of a continuous distribution for the species?
Fig.2 and 3: color coding of individuals (arrows) seems not to be consistent across figures and is not particularly clear. Can an alternative be devised to improve presentation?
Line 239: which is the common pattern in fireflies? Is it WCY as in most insects?
Line 333: abbreviations TRU and RSU are not overly common, please introduce them at first appearance.
Line 338: may be rephrased for clarity. Is CWYshared by all Stenoclaudius genomes and Oculogtyphus? If so, is the Author’s interpretation that a single rearrangement happened in the stem lineage leading to all Stenoclaudius genomes and Oculogtyphus (i.e. not a parallel acquisition)? This makes full sense to me, but it is not totally clear in the manuscript.
Line 345: the proposed translocation model, possibly including a single, and not necessarily 6 duplications, is appropriate to explain the mutation from WCY to CWY. Nevertheless this can hardly explain why the same portion was lost in all 6 repetitions.
I suggest the Authors consider (not necessarily accept, just ‘consider’) also the possibility of a two step process: a) one duplication and random loss (in the common node leading to Stenoclaudius and Oculogtyphus to justify the (shared) inversion; b) multiple duplications in Oculogtyphus, possibly favored by retaining some non coding areas, leading to the presence of repeats in Oculogtyphus.
Fig.5: the role of subpanel 5d is unclear. Is it a duplication of panel 5c? I would have expected 2 Ototretinae clusters (plus one single outlier sequence), and not 3. Furthemore 5c and 5d are virtually identical. Please correct or clarify.
Paragraph 4.3 and elsewhere: two key results stem from the analysis, that are here presented together: a) Oculogtyphus as sister group to Stenoclaudius, and b) poliphyly of asian Ototretinae, with one subgroup marked by Drilaster and one by Oculogtyphus/Stenoclaudius.
In the current presentation, the reader is induced to think Ocylogriphus as member of Ototretinae. Nevertheless, this makes no sense, as Ototretinae themselves are polyphyletic.
I suggest splitting the two issues to discuss the mono/polyphyly of Ototretinae first and the positioning of Ocylogriphus later.
Comments on the Quality of English Language
Some minor corrections are needed, but the content is fully understandable.
Author Response

(The authors gave the same response as above.)
